

# Effects of the visual-feedback-based force platform training with functional electric stimulation on the balance and prevention of falls in older adults: a randomized controlled trial

Zhen Li[1,2], Xiu-Xia Wang[3], Yan-Yi Liang[1], Shu-Yan Chen[2], Jing Sheng[1] and Shao-Jun Ma[1]

[1] Department of Geriatrics, Shanghai Ninth People's Hospital, Shanghai Jiao Tong University School of Medicine, Shanghai, China
[2] Department of Geriatrics, Xinhua Hospital Affiliated to Shanghai Jiao Tong University School of Medicine, Shanghai, China
[3] Department of Plastic and Reconstructive Surgery, Shanghai Ninth People's Hospital, Shanghai Jiao Tong University School of Medicine, Shanghai, China

Corresponding authors
Shu-Yan Chen,
chenshuyan@xinhuamed.com.cn
Jing Sheng, jingsheng60@126.com
Shao-Jun Ma, dabian2017@sina.com

## ABSTRACT

**Background**. Force platform training with functional electric stimulation aimed at improving balance may be effective in fall prevention for older adults. Aim of the study is to evaluate the effects of the visual-feedback-based force platform balance training with functional electric stimulation on balance and fall prevention in older adults.

**Methods**. A single-centre, unblinded, randomized controlled trial was conducted. One hundred and twenty older adults were randomly allocated to two groups: the control group ($n = 60$, one-leg standing balance exercise, 12 min/d) or the intervention group ($n = 60$, force platform training with functional electric stimulation, 12 min/d). The training was provided 15 days a month for 3 months by physical therapists. Medial–lateral and anterior–posterior maximal range of sway with eyes open and closed, the Berg Balance Scale, the Barthel Index, the Falls Efficacy scale-International were assessed at baseline and after the 3-month intervention. A fall diary was kept by each participant during the 6-month follow-up.

**Results**. On comparing the two groups, the intervention group showed significantly decreased ($p < 0.01$) medial–lateral and anterior–posterior maximal range of sway with eyes open and closed. There was significantly higher improvement in the Berg Balance Scale ($p < 0.05$), the Barthel Index ($p < 0.05$) and the Falls Efficacy Scale-International ($p < 0.05$), along with significantly lesser number of injurious fallers ($p < 0.05$), number of fallers ($p < 0.05$), and fall rates ($p < 0.05$) during the 6-month follow-up in the intervention group.

**Conclusion**. This study showed that the visual feedback-based force platform training with functional electric stimulation improved balance and prevented falls in older adults.

## INTRODUCTION

A fall is defined as an event that results in a person coming to rest inadvertently on the ground, or floor, or other lower level (*Hill et al., 2015*). Approximately 30% of community-dwelling adults aged 65 years and older fall at least once a year (*Hirase et al., 2015*). The consequences of falls may be severe, leading to fractures, institutionalization, loss of functional independence, disability, fear of falling, depression and social isolation (*Abreu et al., 2015*). Therefore, prevention of falls among older adults is a crucial public health challenge.

Balance deficits are one of the known risk factors for falls in older adults (*Schmid, Van Puymbroeck & Koceja, 2010*). Previous meta-analyses have shown that multifactorial intervention involving balance training could reduce falls (*Cameron et al., 2012*; *Kannus et al., 2005*). Some traditional balance exercises like Tai Chi as a single intervention has also shown reduced risk of falls in community-dwelling older adults (*Gillespie et al., 2012*). Recently, there is a popular balance training method by instructing participants to stand on a force platform and minimizing the center of pressure (COP) movement (*Dos Anjos, Lemos & Imbiriba, 2016*). Force platform provides a method to quantify and train the ability of maintaining standing position in different circumstances (*Srivastava et al., 2009*). Older adults tend to rely on exteroceptive information and prioritize the vision on postural control. Visual feedback balance training can enhance sensorimotor integration by a recalibration of the sensory systems contributing to balance control (*Hatzitaki et al., 2009*). Force platform balance training combined with visual feedback seems to effect to improve balance control and reduce postural sway in older adults (*Lakhani & Mansfield, 2015*).

The ability of lower extremity muscle to generate adequate force is a fundamental component of maintaining balance and a decrease in muscle strength with aging has been well documented (*Moreland et al., 2004*). Functional electrical stimulation can increase muscle strength, induce changes in muscle fiber composition and capillary system structure, prevent muscle atrophy due to the prolonged immobilization, decreases pain, and increase functional fitness (*Tok et al., 2011*). Functional electrical stimulation has been used to improve balance in adults with stroke and was recently shown some effects on balance in older adults (*Mignardot et al., 2015*).

However, the effects of the visual-feedback-based force platform balance training with functional electric stimulation on the prevention of falls in older adults were unknown. Therefore, the purpose of this study was to evaluate the effects of the visual-feedback-based force platform balance training with functional electric stimulation on the balance and prevention of falls in older adults.

## METHODS

### Participants

This study was conducted at a department of geriatrics in China. Recruitment started in December 2015 and follow-up was completed in January 2017. Inclusion criteria were (1) age 60 and older, (2) ability to walk with or without an assistive device for a minimum of 20 m, (3) ability to see visual feedback from a computer screen, and (4) ability to follow instructions for testing and training (Mini-Mental State Examination, MMSE, >23

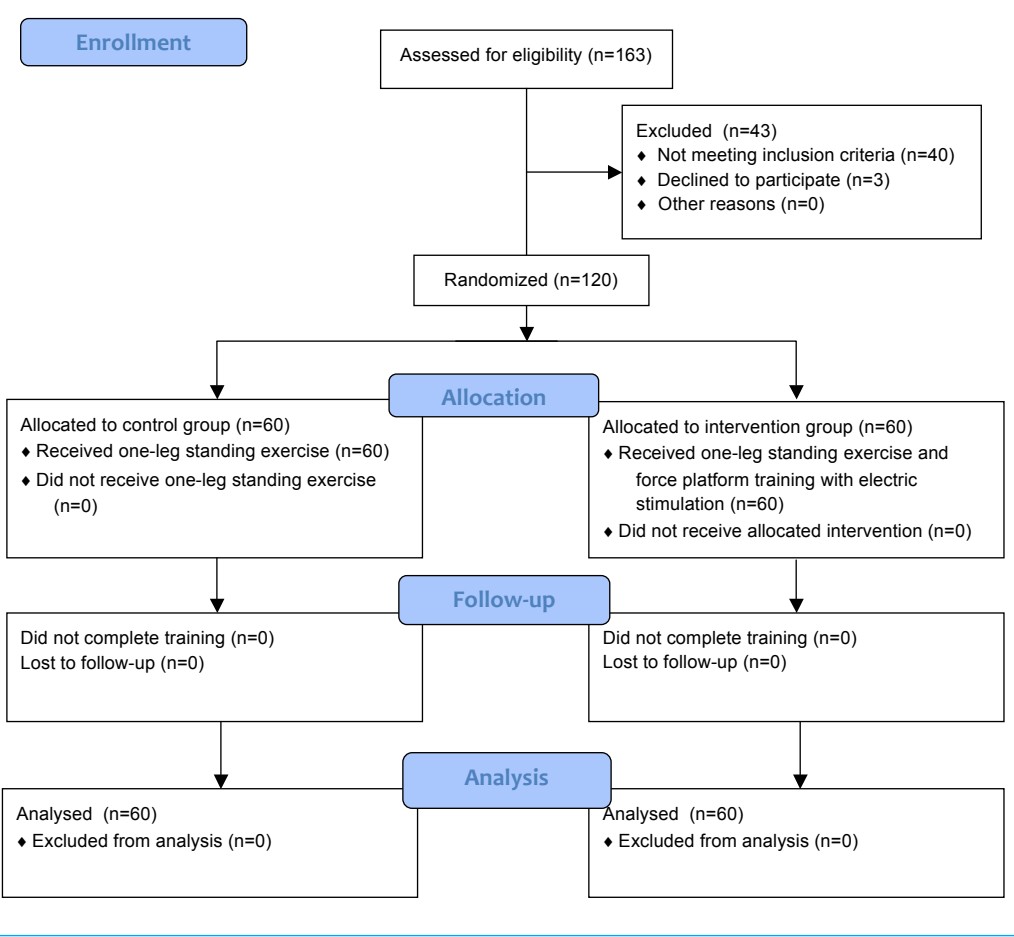

**Figure 1  Flow chart for the study.**

points). Subjects were excluded for any of following reasons: (1) neurological disorders, (2) severe psychiatric disorders (Hamilton Depression Scale, HAMD, >21 point, Hamilton Anxiety Scale, HAMA, >20 point), (3) severe musculoskeletal impairment, (4) terminal cardiovascular problems, and (5) participation in other balance training.

All included participants signed informed consent and they were randomly assigned to the intervention group (IG) or the control group (CG) using computer-generated random numbers. The sequence was concealed until group assignment and baseline measurement was completed. A person unrelated to the study performed the randomization procedure. This study was approved by the Chinese Ethics Committee of Registering Clinical Trials (ChiECRCT-20150041) and was registered at http://www.chictr.org.cn/ (identified: ChiCTR-IOR-16007691). The study design and flow of participants are shown in Fig. 1.

## Intervention

The IG was trained by the computerized force platform with visual feedback (Balance-A, NCC, Shanghai, China) and the functional electrical stimulation with biofeedback (MyoNet-COW, NCC, Shanghai, China).

Participants were instructed to stand barefoot on the force platform in marked foot position (10 cm between heel centers, with an angle of 150 between the long axes of the feet), keeping their arms by their sides. The force platform was connected to a computer and a monitor. The monitor was placed approximately 1 m in front of the participant at eye level. Participants could receive real-time visual feedback of their center of pressure (COP) on the monitor. First, the feedback of their anterior–posterior COP was presented as a fluctuating horizontal line that moved upwards as the COP moved forwards and downwards as the COP moved backwards. A horizontal midline indicating the correct anterior–posterior COP and two horizontal lines indicating 5% skewing of the correct COP were placed on the screen. Participants were told that the feedback line represented the anterior-posterior position of their body and were instructed to keep the line as close to the midline as possible and not exceed the upper and lower boundary lines for 3 min. Then, the feedback of their medial–lateral COP was presented as a fluctuating vertical line. A vertical midline indicating the correct medial–lateral COP and two vertical lines indicating 5% skewing of the correct COP were placed on the screen. Participants were asked to keep the line as close to the midline as possible and to stay in the two vertical boundary lines for 3 min.

After the force platform training, participants were instructed to performed the functional electrical stimulation training in sitting position, with the ankle, knee, and hip joints at 90°. First, Self-adhesive $5 \times 5$ cm electrodes were placed on each leg with the anode with membrane-depolarizing properties positioned over the proximal part of the tibialis anterior muscle about 5 cm below the head of fibula. The cathode was placed on the distal part of the tibialis anterior muscle about 5 cm above the lateral external malleolus of fibula. The starting stimulation intensity was adjusted to elicit maximal contraction without inducing discomfort. Participants were asked to do the ankle dorsiflexion and received electrical stimulation by the device for 3 min (10 s on and 10 s off, rectangular-wave pulsed currents, 50 Hz). Then, the anode was placed along the middorsal line of shank, over both medial and lateral gastrocnemii. The cathode was attached to the superficial aspect of soleus muscle (5 cm in distance from where the two heads of gastrocnemii join the Achilles tendon). Participants were received electrical stimulation when they did the ankle plantarflexion in 3 min.

In addition, both the IG and the CG performed the one-leg standing balance exercise for 12 min per day. The detailed description of one-leg standing balance exercise can be found in a previous article (*Sakamoto et al., 2006*). Participants were not allowed to use a walker or cane during the training. The force platform training with functional electric stimulation and the one-leg standing exercise were provided 15 days a month for 3 months by two different specified physical therapists.

## Outcome measures

*Primary outcome:* Fall rates during 6 months follow-up. Participants using daily fall diaries recorded the number of falls from the day training started until the follow-up ended. The research staff contacted each participant monthly to maximize compliance with the fall diaries. The fall rates are the total number of falls per person in 6 months.

*Secondary outcomes:* The number of injurious fallers and the number of fallers in 6 months follow-up. Falls were classified as injurious if they resulted in a physical injury including bruising, laceration, dislocation, fracture, loss of consciousness, or if the patient reported persistent pain (*Haines et al., 2011*).

COP-based balance parameters were measured using the force platform. Participants were asked to stand still on the force platform for 30 s with their eyes open, then they were asked to close their eyes for 30 s. The following parameters were calculated: maximum range of sway in the medial–lateral and anterior–posterior directions. The maximum range of sway was calculated from the distance between the maximum and minimum COP displacement for each direction. The reliability and validity of COP-based parameters have been tested among older adults (*Li et al., 2016b*).

The Berg Balance Scale (BBS) was used as a functional balance measure. It consists of 14 items that are scored on a scale of 0–4 and is reliable and valid in elderly people. The maximum score is 56, and people with scores below 45 show a risk of falling (*Nick et al., 2016*). The Falls Efficacy Scale-International (FES-I) was a widely accepted tool for accessing the fear of falling. It consists of 16 items that are scored on a scale of 0–4 and the higher scores indicate high concern about falling (*Delbaere et al., 2010*). The ability to perform activities of daily living was assessed using the Barthel Index (BI), which is a valid and reliable tool. The maximum score is 100 and the higher scores represent greater independence (*Ali et al., 2015*). The COP-based parameters, BBS, FES-I, and BI were tested by a therapist initially and after 3 months of balance training. The therapist was blinded to the information about the groups to which the participants belonged.

The sample size estimate was based on extrapolations from our pilot studies and other related promising pilot work (*Li et al., 2016c*; *Mirelman et al., 2013*). Power calculations indicated that a sample size of 120 (60 in each group) would have 80% power to detect a between-group difference of 20% in the fall rates after intervention with a significance level of 0.05 and a dropout rate of 10%.

## Data analysis

Independent-sample t-tests and Chi-square tests were used for baseline comparisons between the IG and CG. Chi-square tests were used to compare sex distributions and fallers in the year before the intervention between the two groups. The group differences in outcome measures were analyzed by two-factor analysis of variance. The between-subject factor was the treatment and the within-subject factor was time. Post hoc Bonferroni tests were performed to compare post-training and baseline measures in each group of patients. Fall rates and the proportion of patients who had falls and injurious falls were compared by Chi-square tests during the 6-month follow-up. Data was analyzed using SPSS (version 21, IBM, New York, USA) and considered significant at $p < 0.05$.

## RESULTS

A flowchart outlining study participation is shown in Fig. 1. A total of 143 inpatients were assessed for eligibility and 120 underwent randomization (60 in each group). No patients dropped out and no training-related adverse events occurred.

**Table 1  Baseline characteristics of the participants.**

| Characteristic | Intervention group ($n = 60$) | Control group ($n = 60$) | *P* value |
|---|---|---|---|
| Age, mean (SD), years | 69.5 (6.0) | 70.6 (5.5) | 0.292 |
| Females, *n* | 29 | 27 | 0.714 |
| BMI, mean (SD), kg/m$^2$ | 24.01 (3.4) | 24.55 (2.7) | 0.335 |
| Systolic BP, mean (SD), mmHg | 146 (13.4) | 148 (13.6) | 0.239 |
| Diastolic BP, mean (SD), mmHg | 83 (10.1) | 81 (8.5) | 0.601 |
| Number of medications, mean (SD) | 5.3 (2.3) | 5.5 (1.9) | 0.696 |
| Fallers in previous 12 months, *n* | 24 | 21 | 0.572 |
| MMSE, mean (SD), score | 26.9 (1.6) | 27.3 (1.7) | 0.342 |
| HAMA, mean (SD), score | 10.4 (3.1) | 11.1 (4.1) | 0.309 |
| HAMD, mean (SD), score | 13.6 (4.1) | 12.8 (3.8) | 0.275 |

Notes.

SD, standard deviation; BMI, body mass index; BP, blood pressure; MMSE, Mini-Mental State Examination; HAMD, Hamilton Depression Scale; HAMA, Hamilton Anxiety Scale.

**Table 2  Effects of force platform balance training with visual feedback on balance.**

| Parameters | Intervention group ($n = 60$) | | | Control group ($n = 60$) | | | |
|---|---|---|---|---|---|---|---|
| | Baseline | Post-training | $p^a$ | Baseline | Post-training | $p^a$ | $p^b$ |
| BI | 93.1 (2.6) | 96.3 (3.2) | <0.001 | 93.9 (2.1) | 95.1 (2.7) | 0.005 | 0.026 |
| BBS | 44.5 (5.1) | 50.1 (3.1) | <0.001 | 45.2 (4.7) | 48.7 (3.7) | <0.001 | 0.027 |
| FES-I | 31.4 (8.2) | 22.9 (7.5) | <0.001 | 32.1 (8.6) | 26.4 (8.1) | <0.001 | 0.015 |
| Eyes open | | | | | | | |
| $M_{ML}$, cm | 3.81 (1.05) | 2.14 (0.34) | <0.001 | 3.57 (0.56) | 2.74 (0.38) | <0.001 | 0.001 |
| $M_{AP}$, cm | 2.63 (0.52) | 1.25 (0.65) | <0.001 | 2.45 (0.75) | 1.85 (0.15) | <0.001 | <0.001 |
| Eyes closed | | | | | | | |
| $M_{ML}$, cm | 1.86 (0.79) | 0.83 (0.25) | <0.001 | 1.56 (0.94) | 1.03 (0.47) | <0.001 | 0.004 |
| $M_{AP}$, cm | 2.49 (0.87) | 1.86 (0.44) | <0.001 | 2.54 (0.61) | 2.16 (0.73) | 0.002 | 0.007 |

Notes.

Data are mean (SD).

[a] *p* values for within group comparisons ($p < 0.05$).

[b] *p* values for treatment × time interaction ($p < 0.05$).

BI, Barthel Index; BBS, Berg Balance Scale; FES-I, Falls Efficacy Scale-International; $M_{ML}$, medial–lateral maximal range of sway; $M_{AP}$, anterior–posterior maximal range of sway.

The baseline characteristics of the participants are summarized in Table 1. There were no significant differences between the two groups in terms of age, sex, BMI, blood pressure, number of medications, fallers in the previous year, MMSE, HAMD, and HAMA ($p > 0.05$).

The results of baseline and post-training balance assessment are given in Table 2. No significant difference was found between the two groups at baseline of BI, BBS, FES-I, and COP-based parameters ($p > 0.05$).

The BI score and BBS score were both increased in the IG (from 93.1 ± 2.6 to 96.3 ± 3.2, from 44.5 ± 5.1 to 50.1 ± 3.1) and CG (from 93.9 ± 2.1 to 95.1 ± 2.7, from 45.2 ± 4.7 to 48.7 ± 3.7) after training and the BBS scores in both groups were increased more than

**Table 3  Effects of force platform balance training with visual feedback on fall.**

|  | IG ($n = 60$) | CG ($n = 60$) | P value |
|---|---|---|---|
| Injurious fallers, $n$ | 5 | 13 | 0.041 |
| Fallers, $n$ | 8 | 17 | 0.043 |
| Fall rates, % | 33.3 | 53.3 | 0.027 |

**Notes.**

P values are given for difference between groups.

IG, intervention group; CG, control group.

45. There were significant differences between two groups ($p < 0.05$). The FES-I score in the IG (from $31.4 \pm 8.2$ to $22.9 \pm 7.5$) and in the CG (from $32.1 \pm 8.6$ to $26.8 \pm 8.1$) also reduced and there were significant differences between two groups ($p < 0.05$). The FES-I score in the IG $< 23$ indicated low concern about falling.

The maximal range of sway in the medial–lateral and anterior–posterior directions were reduced in both the groups with eyes open and eyes closed conditions at the end of the 3 months training. The medial–lateral maximal range of sway reduced from $3.81 \pm 1.05$ to $2.14 \pm 0.34$ and the anterior–posterior maximal range of sway reduced from $2.63 \pm 0.52$ to $1.25 \pm 0.65$ in eyes open in the IG. In the CG they were reduced from $3.57 \pm 0.56$ to $2.74 \pm 0.38$ and from $2.45 \pm 0.75$ to $1.85 \pm 0.15$. The medial–lateral maximal range of sway reduced from $1.86 \pm 0.79$ to $0.83 \pm 0.25$ and the anterior–posterior maximal range of sway reduced from $2.49 \pm 0.87$ to $1.86 \pm 0.54$ in eyes closed in the IG. In the CG they were reduced from $1.56 \pm 0.94$ to $1.03 \pm 0.47$ and from $2.54 \pm 0.61$ to $2.16 \pm 0.73$. There was significant difference between two groups ($p < 0.01$).

Table 3 shows the number of injurious fallers, the number of fallers, and fall rates in 6 months follow-up. It was found that the number of injurious fallers during the 6 months follow-up in the IG (5) was less than the CG (13) and the difference between the two groups was statistically significant ($p < 0.05$). The number of fallers in the IG (8) was also less than the CG (17) during the 6 months follow-up and there were significant differences between two groups ($p < 0.05$). The fall rates in IG were 33.3% and in CG were 53.3% during the 6 months follow-up. There were significant differences between two groups ($p < 0.05$).

## DISCUSSION

The results of this randomized, controlled study showed that the visual-feedback-based force platform training with functional electric stimulation can significantly improve balance and prevent falls in older adults. The results also showed a positive effect of the training on activities of daily living and the reduction of falling fear. To the best of our knowledge, this is the first study using the visual-feedback-based force platform training with functional electric stimulation to test the effects on the balance and prevention of falls in older adults.

The maximal range of sway in the medial–lateral and anterior–posterior directions were reduced significantly in the IG indicated that participants could control their center of pressure more stably and accurately. This meant these participants improved the ability to maintain the standing position in static circumstances and the ability to maintain standing

while experiencing internally produced perturbations associated with movements of their extremities (*Barclay-Goddard et al., 2004*). The learning of balance skills was facilitated by the use of visual feedback. Visual information on how the center of pressure is situated and on how it moves during different tasks serves as a tool to improve the volitional postural control. This result was consistent with the motor learning theory that emphasized the role of feedback in the learning of motor skills (*Sihvonen, Sipila & Era, 2004*). The results of the BBS showed improvement in the performance of functional balance after training. These indicated that the participants improved their ability of balance when they did some real-life meaningful activities such as transfers and stair climbing after training. These meant that the positive effect of training on balance was founded in both functional balance measures as well as laboratory measures. Some previous studies also used other scales to test the effect of the visual-feedback-based balance training on balance. For example, *Schwenk et al. (2014)* tested thirty-three older adults using visual feedback technique and founded the Timed-up-and-go improved significantly in the IG.

This study evaluated effect of training not only on balance but also on the ability to perform the activities of daily living. The results of the BI showed that the IG significantly improved the ability to perform activities of daily living after training. This means they can perform everyday actions more safely and independently. *Srivastava et al. (2009)* also reported that the BI scores were improved in forty-five stroke survivors after four weeks visual-feedback-based force platform training. Meanwhile, the results of the FES-I showed the less concerned about falls in the IG compared with the CG after training. The consequences of fear of falling include falling, restriction or avoidance of daily activities, loss of independence, reduction in social activity, depression and a reduction in quality of life (*Kendrick et al., 2014*). The reduction in fear of falling improved not only the confidence regarding falls but also the confidence in social activities. This indicated that this training might have positive effect on life quality of older adults.

It should be noted the BBS score for the IG group was raised to a level that have less risk to fall as previously mentioned in the Methods section (*Nick et al., 2016*). The FES-I score in the IG also reached a level that has low concern about falling after three months training (*Delbaere et al., 2010*). This study found that the fallers and fall rates were both decreased significantly in the IG during six months follow-up. This indicated the visual-feedback-based force platform training with functional electric stimulation could prevent falls in older adults. Previous studies have tested the effect of visual-feedback-based force platform training as a single intervention for fall prevention in older adults. Although these studies showed that the force platform training could improve balance, they found that the force platform training singly might have not significant effect on prevention of falls. For example, *Wolf et al. (2003)* found no significant difference between training group and education group in the rate of falls after 15 weeks training and 4-month follow-up in community-dwelling older adults. *Sihvonen et al. (2004)* found no significant difference in number of fallers between two groups after 4 weeks training and 1-year follow-up in 27 older women living in residential care. They only found the recurrent falls were significantly decreased in the IG compared to the CG. This maybe because visual-feedback-based force platform training emphasizes more on the improvement of the organization of sensory

information related to postural control and has no significant effects on ankle muscles function (*Hagedorn & Holm, 2010*). But humans need to generate appropriate torques at the ankle joint to control body sway while standing upright and the decrease in the maximal strength of ankle muscles are considered a main cause of postural instability (*Cattagni et al., 2014*). In this study, the tibialis anterior muscle and triceps surae muscle were trained by the functional electrical stimulation. It has been proved the functional electrical stimulation can increase muscle function of lower limbs in older adults (*Maggioni et al., 2010*). A meta-analysis included 24 trials showed that electrical stimulation improved the functional motor ability and normality of movement in patients with stroke (*Pomeroy et al., 2006*). In addition, this study used randomization and set strict exclusion criteria to minimize other risk factors' affection on falls, for example, HAMD and HAMA showed no significant difference between the two groups at the baseline. Besides, the IG in this study was given a total of 13.5 h training which was more than twice the training duration of previous studies (*Sihvonen et al., 2004*). So the visual-feedback-based force platform training with functional electric stimulation showed the positive effects on the prevention of falls in older adults in this study. Furthermore, the injurious fallers were also decrease significantly in the IG during six months follow-up. The reasons for reduction of the injurious fallers were not only the decreased fallers number but also the improved ability to restore balance in response to obstacles and sudden perturbations in the walking as the improved muscle function of low limbs (*Pomeroy et al., 2006*).

There were limitations in the present study. First of all, a randomized controlled trial with a double-blind study design was not possible because of the nature of the intervention. So the study designed some methods to reduce the bias. For instance, the different exercises were conducted by different physical therapists and data were collected by a specified person who was blinded to the grouping condition. Secondly, this study did not use other functional balance measures (such as Timed Up-and-Go) and muscle strength measures. Meanwhile, Virtual Reality (VR) as a novel technology can provide visual, vestibular, and somatosensory feedback (*Li et al., 2016a*). It can also provide a mimic environment that the participant thinks real and environment factor is an independent risk factor of falls (*Collado-Mateo et al., 2017*; *Fu et al., 2015*). So future study should add VR and more measures into the training to get more enhanced effects and evidences. Finally, this study did not recruit some frail older adults because of the strict inclusion and exclusion criteria. So future study needed to evaluate the effect of the training on the frail older adults.

## CONCLUSIONS

This study showed that the visual-feedback-based force platform training with functional electric stimulation could improve balance and activities of daily living in older adults, thereby preventing falls. The visual-feedback-based force platform training with functional electric stimulation was feasible with good adherence and few adverse events. A future larger study is needed to test the effects of the visual-feedback-based force platform training with functional electric stimulation.

### Funding
This study was funded by Science and Technology Commission of Shanghai Municipality (No. 13DZ1941606). The funders had no role in study design, data collection and analysis, decision to publish, or preparation of the manuscript.

### Grant Disclosures
The following grant information was disclosed by the authors:
Science and Technology Commission of Shanghai Municipality: 13DZ1941606.

### Competing Interests
The authors declare there are no competing interests.

### Author Contributions
- Zhen Li wrote the paper, prepared figures and/or tables, reviewed drafts of the paper.
- Xiu-Xia Wang analyzed the data, contributed reagents/materials/analysis tools.
- Yan-Yi Liang conceived and designed the experiments, reviewed drafts of the paper.
- Shao-Jun Ma and Shu-Yan Chen performed the experiments.
- Jing Sheng conceived and designed the experiments.

### Clinical Trial Ethics
The following information was supplied relating to ethical approvals (i.e., approving body and any reference numbers):

This study was approved by the Chinese Ethics Committee of Registering Clinical Trials. Approval No. ChiECRCT-20150041.

### Data Availability
The raw data has been provided as Supplemental File.

### Clinical Trial Registration
The following information was supplied regarding Clinical Trial registration:

ChiCTR-IOR-16007691.

### Supplemental Information
Supplemental information for this article can be found online at http://dx.doi.org/10.7717/peerj.4244#supplemental-information.

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
