# Peer review of "Effects of the visual-feedback-based force platform training with functional electric stimulation on the balance and prevention of falls in older adults: a randomized controlled trial"

_PeerJ, doi:10.7717/peerj.4244_

## Round 0.1 · original submission · Minor Revisions

The two reviewers and I are impressed with most aspects of your study design and the way in which the manuscript is written. We therefore are happy to recommend you are given the opportunity to respond to these comments.

·

Basic reporting

While the manuscript adheres to a professional structure, there needs to be more precision with professional English, most notably in the introduction and discussion.
For example – linking words related to a fall or falls: line 35 (insert A before fall), line 41 (change to balance deficits are one of the known risk factors for falls), line 45 (insert to before stand), line 46-48 (needs re-wording), line 229 (reached instead of reach).

There is sufficient background within the introduction but further context could be included in the discussion regarding electrical stimulation.

Experimental design

Methods explained in appropriate detail allowing for replication.
It should be made clearer that the control group only undertook single leg balance training and did not receive further training or electrical stimulation.
Pictures of standing position/visual feedback provided may help to illustrate intervention protocols.
Effect sizes should also be included in table 2.

Further information on falls and associated injuries would be also be valuable.

Validity of the findings

Overall the data is robust and controlled.

Were improvements in balance and reductions in falls due to the balance training or the electrical stimulation? Or a combination?I think the authors might be overstating the potential impact of electrical stimulation without strong links/justification. As improvements in strength were not measured it is difficult to say how much of an impact the electrical stimulation training had. It is more likely that the visual feedback balance training (together with the single leg balance training) provided most of the improvements in balance and associated reduction in falls. Whilst there are links to improvements in strength and possibly function with electrical stimulation, these associations have not been strongly argued.The authors could expand on the reasons for this improvement (associated with balance training and visual feedback) in more detail as it is a very interesting and important finding.


In the conclusion – few adverse events (line 277). Does this mean there were some adverse events during training/testing – please clarify.

Additional comments

I commend the authors on conducting this well powered balance intervention in older adults. The force platform based visual feedback is certainly an interesting area that has great potential for improving balance and potentially reducing falls in older adults.

Reviewer 2 ·

Basic reporting

• Manuscript was sent to me UNBLINDED.
• Raw data was provided.
• Manuscript was very clearly written and easy to follow

Experimental design

• Research question was well defined, relevant and meaningful. The results fill the gap of how platform and electrical stimulus training can improved measures of balance and reduce falls.
• Methods described in detail
• Reported with rigorous RCT methods

Validity of the findings

• This research seems to be valid for bridging the gap between platform and electrical stimulus training. On a side note, it would be interesting to see how effective each individual intervention (platform, stimulation & single leg balance) was.
• Data is robust, statistically sound and controlled.
• Conclusions were well stated

Additional comments

• Overall the paper is very well written, easy to follow and methodologically sound.
• I would speculate that perhaps a Repeated Measures ANOVA may be a more appropriate statistical method to use, but the description of the two-way factorial ANOVA sounds sufficient.

---

## Round 0.2 · accepted · Accept

I am happy to say that you have addressed all of the relatively minor concerns we had with the initial version of the manuscript and as such we are happy to accept this manuscript for publication in PeerJ.

·

Basic reporting

No comment

Experimental design

No comment

Validity of the findings

The authors have clarified their findings in the results and discussion. This clarification was aided by the edits made to the methods.

Additional comments

Thank you for making the suggested changes to the manuscript. The paper now flows better and the discussion and conclusions are now better supported.
Congratulations on completing such an interesting and well powered study.